# Pro-social behavior in rats is modulated by social experience

**Inbal Ben-Ami Bartal[1], David A Rodgers[1], Maria Sol Bernardez Sarria[1], Jean Decety[2,3,4], Peggy Mason[1,4]***

[1]Department of Neurobiology, University of Chicago, Chicago, United States; [2]Department of Psychology, University of Chicago, Chicago, United States; [3]Department of Psychiatry and Behavioral Neuroscience, University of Chicago, Chicago, United States; [4]Committee on Neurobiology, University of Chicago, Chicago, United States

**Abstract** In mammals, helping is preferentially provided to members of one's own group. Yet, it remains unclear how social experience shapes pro-social motivation. We found that rats helped trapped strangers by releasing them from a restrainer, just as they did cagemates. However, rats did not help strangers of a different strain, unless previously housed with the trapped rat. Moreover, pair-housing with one rat of a different strain prompted rats to help strangers of that strain, evidence that rats expand pro-social motivation from one individual to phenotypically similar others. To test if genetic relatedness alone can motivate helping, rats were fostered from birth with another strain and were not exposed to their own strain. As adults, fostered rats helped strangers of the fostering strain but not rats of their own strain. Thus, strain familiarity, even to one's own strain, is required for the expression of pro-social behavior.

## Introduction

Pro-social behavior comprises actions that improve the well-being of others (*Eisenberg et al., 1989*) and is found widely in the animal world (*Owens and Owens, 1984*; *Wilkinson, 1984*; *Lee, 1987*; *Heinrich and Marzluff, 1995*; *Warneken and Tomasello, 2006*; *Nowbahari et al., 2009*; *Yamamoto et al., 2012*; *Baden et al., 2013*; *Clay and de Waal, 2013*; *Hatchwell et al., 2013*). Pro-sociality within a group promotes individual survival and reproduction (*Hamilton, 1984*; *Preston and de Waal, 2002*; *Decety and Svetlova, 2012*). On the other hand, acting pro-socially towards individuals from other groups may not be adaptive, as other groups often compete for valuable and limited resources. Thus animals can be motivated to act pro-socially or aggressively depending on the social context.

In humans, pro-social behavior is modulated by the degree of affiliation and is extended preferentially towards in-group members and less often toward unaffiliated others (*Hornstein, 1978*; *Cialdini et al., 1997*; *Preston and de Waal, 2002*; *Levine et al., 2005*; *Sturmer et al., 2006*; *Lamm et al., 2010*; *Echols and Correll, 2012*). Yet humans can and often do act pro-socially towards strangers (*Batson et al., 2005*). This human capacity to help unfamiliar individuals is often viewed as a cognitively complex behavior that depends on high cognitive capacities and cultural transmission (*Levine et al., 2001*).

Rodents have emerged as a valuable model system for social behavior and communication (*Decety and Svetlova, 2012*; *Mogil, 2012*; *Panksepp and Panksepp, 2013*; *Preston, 2013*). Rodents manifest emotional contagion (*Langford et al., 2006*; *Chen et al., 2009*; *Jeon et al., 2010*; *Knapska et al., 2010*; *Panksepp and Lahvis, 2011*; *Akyazi and Eraslan, 2014*; *Atsak et al., 2011*), cooperation (*Rutte and Taborsky, 2007*; *Viana et al., 2010*; *Tsoory et al., 2012*), and helping (*Ben-Ami Bartal et al., 2011*; *Church, 1959*; *Rice and Gainer, 1962*). We previously found that Sprague-Dawley (SD) rats learn to release SD cagemates trapped in a restrainer (*Ben-Ami Bartal et al., 2011*). Here, we set

*For correspondence: pmason@uchicago.edu

**eLife digest** Humans help family members and friends under circumstances where they may not help strangers. However, they also help complete strangers through both direct actions, such as helping someone who has stumbled, and indirect actions, such as giving to charity. Ben-Ami Bartal et al. have now explored the biological basis of such socially selective helping by testing whether rats help strangers, and if so, under what circumstances.

In the experiments a free rat was exposed to another rat trapped inside a plastic tube with an outward-facing door for 12 one-hour sessions. When tested with a cagemate trapped inside the tube, most free rats learned within a few days to release the trapped rat by opening the door. Ben-Ami Bartal et al. then exposed the free rats to strangers they had never met or seen before. Remarkably the rats consistently released the trapped stranger, acting toward strangers just as they had acted toward familiar cagemates. This result suggested that individual familiarity is not required for helping to occur.

To test the limits of rat benevolence, Ben-Ami Bartal et al. tested free rats (always white albino rats) with trapped rats from a different outbred strain (black-hooded rats). The rats helped cagemates of a different strain but not strangers of a different strain. These results could be explained by a requirement for strain familiarity or individual familiarity. To distinguish between these possibilities, albino rats were housed for 2 weeks with a rat of a different strain, and then re-housed with another albino rat before being tested with a trapped rat belonging to a different strain. Consistent with a requirement for strain but not individual familiarity, the free rats now helped stranger rats from the different, but now familiar, strain.

To explore if there is any role for genetics or relatedness in socially selective helping, Ben-Ami Bartal et al. tested whether rats will help strangers of their own strain based on genetic relatedness alone. To do this albino pups were transferred to litters of a different strain on the day they were born, and never saw or interacted with another albino rat until testing. Remarkably, the albino rats helped strangers from the different strain that they were raised with, but they did not help strangers of their own strain because this strain was unfamiliar to them. The fact that the motivation to help other rats has its origins in social interactions rather than genetics provides the flexibility that is needed to navigate their way through social environments that often change unexpectedly.

out to investigate how previous social experience acquired during development and adulthood influences helping behavior. First, we tested whether rats will help unfamiliar individuals, strangers. Rats showed helping behavior equally towards cagemates and strangers, if the strangers were of their own strain. However, rats did not help strangers of an unfamiliar strain, suggesting that helping in rats may be innately biased towards the helper's own strain. Yet, further experiments demonstrated that a short period of pair-housing with a rat of the unfamiliar strain was sufficient to motivate helping for that individual. Moreover, rats that had previously lived with a rat of a different strain were as motivated to help strangers of that strain as they were to help strangers of their own strain. This finding demonstrates that rats choose to help others depending on the social context, and extend pro-social motivation beyond individual identity, to groups defined by strain. Finally, we sought to determine if genetic relatedness is at all capable of influencing pro-social motivation towards an unfamiliar rat. We found that rats that were fostered from birth with another strain did not help strangers of their own strain as adults. Fostered rats only acted pro-socially towards rats of the foster strain, demonstrating that genetic relatedness alone is not capable of producing pro-social motivation.

In each hour-long session, a free rat of the SD strain was placed in an arena containing another rat trapped inside a centrally located restrainer. (A rodent stock, colloquially referred to as an outbred strain, is a colony of conspecifics derived from a small group of founder animals. Individuals are not genetically identical. Here we refer to stocks as strains.) The door to the restrainer could only be opened from the outside and thus only by the free rat. Door-opening by the free rat led to the trapped rat's release from the restrainer. If a free rat failed to open the door within 40 min, the experimenter opened it halfway, allowing the trapped rat to exit the restrainer. Only door-openings resulting from the free rat's action, during the first 40 min of each session, were counted as such. Regardless of which rat opened the restrainer door and whether that occurred before or after halfway opening,

both rats remained in the arena for the entire hour of every session. All rats were males and all free rats were from the SD strain. Sessions were repeated for 12 days.

## Results

To determine whether rats help unfamiliar individuals, free SD rats were placed with trapped SD rats that were either strangers (n = 12, SD stranger condition) or cagemates (n = 8, SD cagemate condition) in the helping behavior test described above. Cagemates were pair-housed for 2 weeks prior to the experiment. Strangers were SD males from a different cohort, born in the same week but on a different day as the test rats. On each day of testing, a different stranger, to whom the free rat had no prior exposure, was trapped in the restrainer. Most rats in both SD cagemate (6/8, 75%) and SD stranger (10/12, 83%) conditions acted pro-socially, learning to release the trapped rat, and becoming openers (*Figure 1*; see Methods for opener definition). Rats in the two conditions were similarly active as measured by velocity (two-tailed Student's *t* test, p>0.05), spent similar amount of time near the trapped rat (two-tailed Student's *t* test, p>0.05), and began to open the door on around the same day (4.0 ± 1.1 days for trapped cagemates; 3.7 ± 0.8 for trapped strangers, mean ± SEM). Thus, rats were as motivated to help strangers as they were to help cagemates, showing that individual familiarity is not required for pro-social behavior in rats.

The results above indicate that rats are motivated to help unfamiliar rats of their own strain. To determine if rats extend help to rats from a different strain, free SD rats were tested in the helping behavior test with trapped rats of the black-caped Long-Evans (LE) strain, which were either cagemates (n = 7, LE cagemate condition) or strangers (n = 16, LE stranger condition). SD rats did not release LE strangers with only a minority becoming openers (4/16, 25% openers; *Figure 2*; *Video 1*). In contrast, most SD rats in the LE cagemate condition became openers (5/7, 71%; χ2, p=0.04), learning to open the restrainer on average on day 3.8 ± 1.7 (mean ± SEM, *Figure 2*). Thus, rats help strangers of their own strain but not strangers of a different strain, suggestive of an in-strain bias for pro-social behavior. Moreover, short-term, paired-housing with a rat from a different strain is sufficient to motivate helping for that individual rat.

We then examined whether SD rats that were familiar with one LE individual would be motivated to help LE strangers. SD rats were pair-housed with an LE cagemate for 2 weeks, then re-housed with an SD rat, and a week later, tested in the helping behavior test with trapped LE strangers (n = 12, LE familiar condition; *Figure 2*). In contrast to SD rats with no exposure to the LE strain, most SD rats familiar with one LE individual became openers for LE strangers (8/12, 67%; χ2, p=0.03). They learned to open the door on average on day 4.0 ± 1.0 despite a lack of individual familiarity with the trapped LE rat (*Figure 2*). This experiment shows that social experience with one individual rat of a different strain is sufficient to motivate helping towards unfamiliar members of that strain. This provides further evidence that individual familiarity is not required for helping.

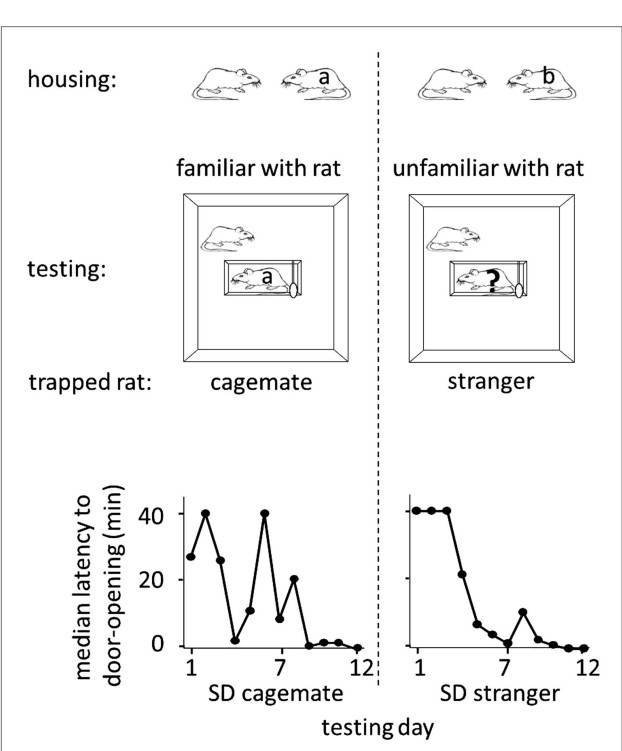

**Figure 1**. Rats were as motivated to help strangers of the same strain (SD stranger; right) as they were to help cagemates (SD cagemate; left). In both experimental conditions (diagrammed at top), SD rats were housed with another SD rat. However in the cagemate condition, the cagemate served as the trapped rat ('a') whereas in the stranger condition, the trapped rat was a stranger ('?') and not the cagemate ('b'). Across the days of testing, the median latency to door-opening (bottom) decreased for both conditions.

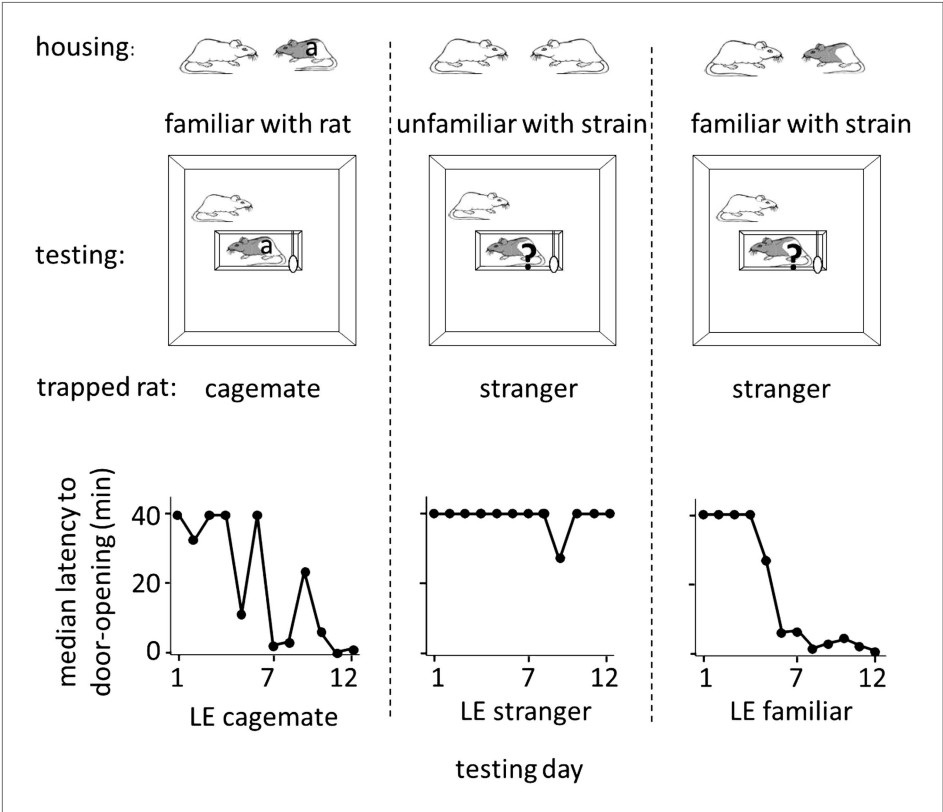

**Figure 2**. Rats of the SD strain helped trapped rats of the LE strain only if they were familiar with an LE individual. Three experimental conditions are diagrammed at top. The free rat was always from the SD strain and was housed with a cagemate, denoted at top, from either the SD (white) or LE (black and white) strain. The free rat was then tested with an LE rat that was either the cagemate ('a') or a stranger ('?'). Note that in the LE familiar condition (right), rats had previously housed with an LE rat (illustrated) but were housed with an SD rat (not illustrated) at the time of testing. Across the days of testing, the median latency to door-opening (bottom) decreased for SD rats tested with trapped LE cagemates (left), but not those tested with LE strangers (middle). Like rats in the LE cagemate condtion, rats in the LE familiar condition (right) also became openers.

Finally, to determine if helping behavior is at all influenced by the strain of the trapped rat (SD, LE), or if strain bias in helping is entirely due to strain familiarity (familiar, unfamiliar), SD rats were fostered and raised with LE rats from birth, in an environment that effectively prevented exposure to others of their own strain (fostered conditions, **Figure 3**). If rats are innately motivated to help rats of their own strain, then SD rats raised exclusively with LE rats (fostered) should act pro-socially towards other SDs. In contrast, if strain familiarity is the only determinant of pro-social behavior, these rats should not help their own kind. At 2 months of age, fostered rats were tested with trapped rats that were SD strangers (n = 8, fostered+SD) or LE strangers (n = 8, fostered+LE). Fostered SD rats did not help trapped SD strangers (**Figure 4**; **Video 2**). Only 1 of 8 fostered SD rats (12.5%) became an opener, establishing that rats are not innately motivated to help their own strain. In contrast, fostered rats did help LE strangers (5/8, 62.5%; χ2, p=0.04) as expected from the pro-social influence of strain familiarity. Thus, strain familiarity, even to one's own strain, is required for the expression of helping behavior.

The finding that fostered rats did not help SD strangers indicates that a rat's familiarity with himself is insufficient to motivate helping for individuals of his strain. Instead, social interaction with another rat, such as that occurring while two rats live together, is critical to shaping pro-social motivation. Moreover, since rats tested with strangers from unfamiliar strains did not become openers across the days of testing, the exposure and interactions afforded by testing sessions appear insufficient to produce pro-social motivation.

Fostered rats were noticeably more anxious than naturally reared rats. In open-field testing conducted prior to the experiment, fostered rats spent more time in the corners of the arena than rats

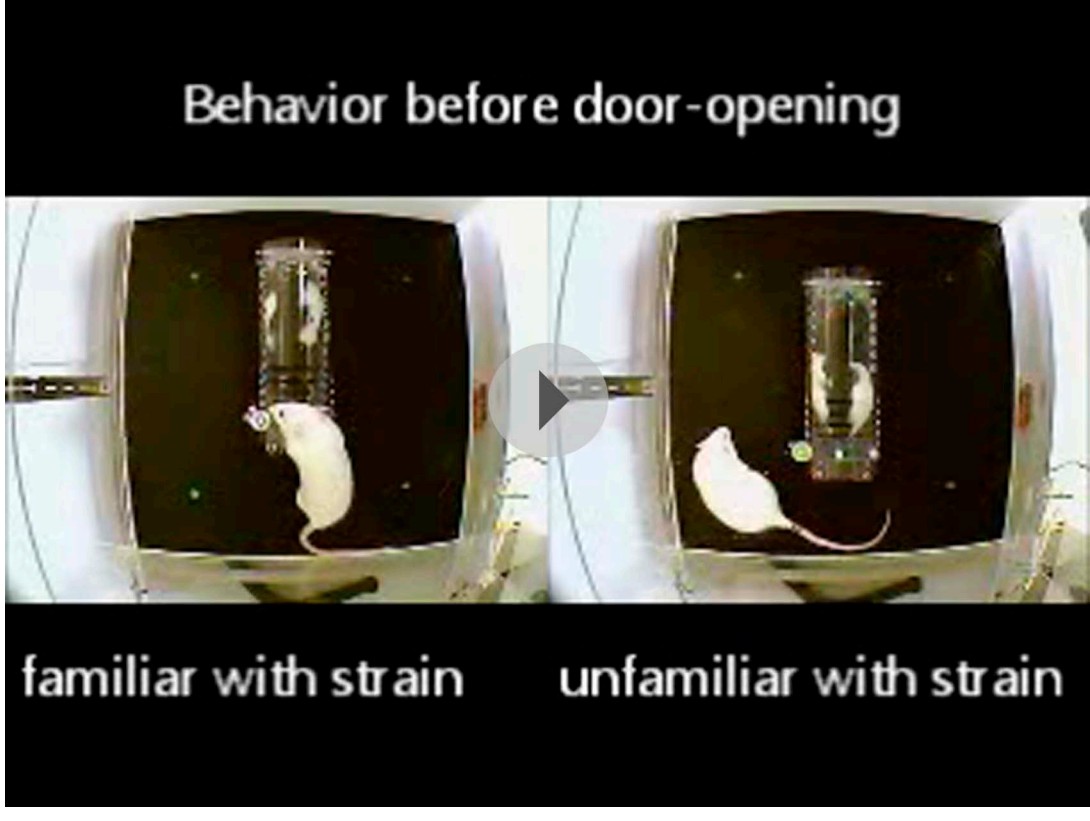

**Video 1**. Rats help rats of a familiar strain. Rats that are familiar with at least one LE rat show interest in, open the restrainer door for, and rarely fight with trapped LE rats. Rats that are unfamiliar with LE rats show little interest in, do not help, and often fight with an LE rat.

in other conditions (33.1 ± 2.7 s/min for fostered rats; 25.0 ± 1.9 s/min for naturally reared rats; files with open field data for SD cagemate and SD strangers were corrupted and therefore not included; two-tailed Student's *t* test, p<0.02; *Figure 5A*). Moreover, fostered rats that opened the restrainer began to do so significantly later (day 8.0 ± 1.1 for fostered rats; 4.5 ± 0.5 for all other rats; two-tailed Student's *t* test, p<0.01; *Figures 1–2, 4B*). Fostered rats were tested for an extra day, confirming that opening behavior persisted. These data suggest that fostering rats in a different-strain environment is associated with increased anxiety. A same-strain, cross-fostering control condition is needed to distinguish whether strain, fostering, or both drive this effect.

Across all conditions, the free rat's familiarity with the strain of the trapped rat is the common determinant of whether helping behavior occurred (*Figure 6A*). Most rats in familiar strain conditions became openers (33/47, 74%) whereas only 21% (5/24) of rats in unfamiliar strain conditions did (χ2, p<0.001). Rats tested with a familiar strain experienced opening the restrainer door as rewarding, as they typically opened it on consecutive days (*Figure 6B–C*). In contrast, rats tested with unfamiliar strains rarely opened on consecutive days, suggesting that opening was not rewarding for these animals. As open field performance and alarm calls were not different for familiar and unfamiliar strain conditions (*Figure 5C*), it is unlikely that trait anxiety or the trapped rat's distress accounts for the results.

When tested with familiar strains, helping behavior was demonstrated equally for strangers and cagemates. Yet, the underlying affective response might differ in these conditions. In support of this idea, rats displayed a different movement pattern when tested with a trapped cagemate or stranger. Rats tested with a trapped cagemate were significantly more active prior to door-opening than rats tested with a stranger (MMA, p<0.05, *Figure 7A*). However, there was no difference in the distance from the restrainer in these conditions (*Figure 7B*). These results demonstrate that rats were equally motivated to help, but not equally aroused by, a trapped cagemate and stranger.

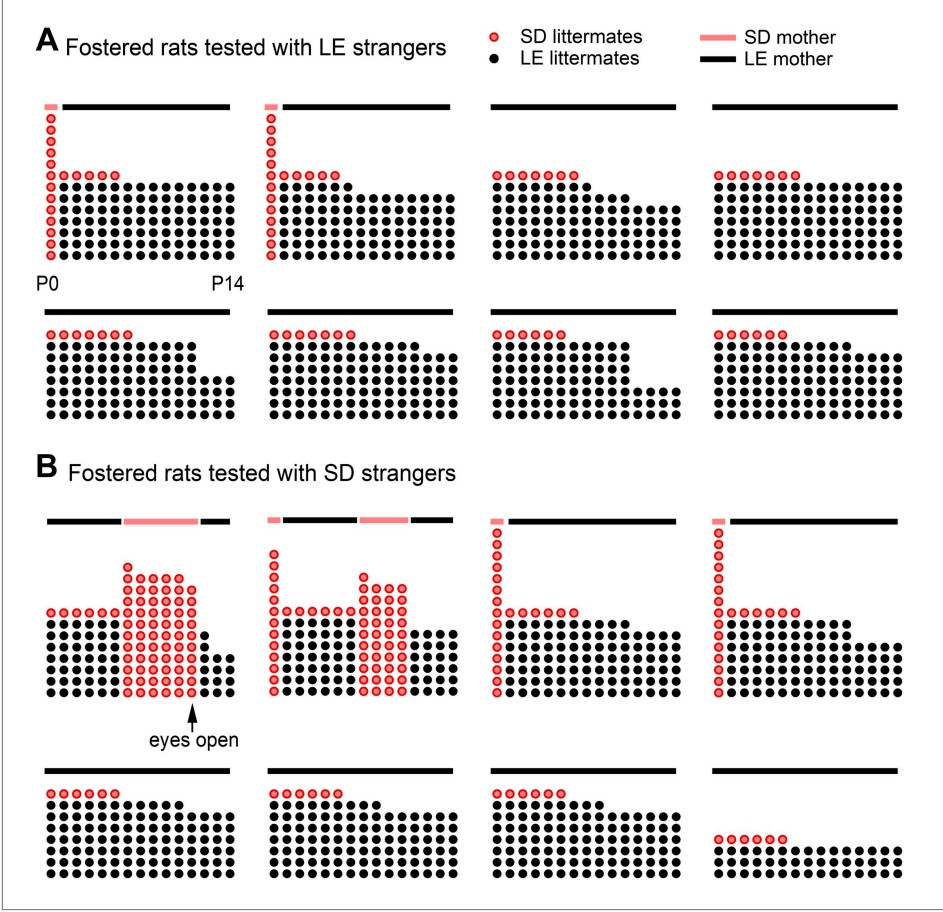

**Figure 3**. Fostered SD rats were minimally exposed to other SD rats from birth. This diagram indicates how many LE pups (black dots) and SD pups (red dots) were present in each litter during postnatal (P) days 0–14 (x-axis) for rats in the fostered+LE (**A**) and fostered+SD (**B**) conditions (see *Figure 4*). On P0–P1, two SD pups (n = 32) were transferred into each LE litter (n = 16). If both pups survived one of the two SD pups was removed at P6, leaving a single SD pup with an LE dam and LE littermates. As red dots indicate other SD pups, the number of red dots is representative of the amount of exposure each fostered SD rat had to other SD pups during their lives. Individual rats are arranged from the most exposure to SD rats (upper left in each group) to the least exposure (bottom right). Following P11, no fostered rat was exposed to other SD rats until testing. In two cases (both in the fostered+SD condition illustrated in **B**), the SD pup in the LE litter died at a later date (P11, P12). In these cases, an SD male who had been removed from an LE litter 5 or 7 days before was then re-added to the LE litter. Note that only one animal (upper left in **B**) was exposed to SD rats with his eyes open; this rat proved to be the only door-opener in the fostered+SD condition. Lines above the dots represent the strain of the dam (red: SD; black: LE).

Door-opening behavior overlapped greatly, but not completely, with strain familiarity. Across all conditions, rats who became openers demonstrated an affiliative behavioral pattern. Prior to door-opening, openers spent more time around the closed restrainer than did non-openers (two-tailed Student's t test, p<0.01; *Figure 8A*; *Video 1*). Following the trapped rat's exit from the restrainer, significantly less fights were observed for openers than for non-openers (two-tailed Student's t test, p<0.01; *Figure 8A*). Baseline anxiety level was assessed by the time spent in the corners of the arena during open-field testing. Only for openers, baseline anxiety was positively correlated with the first day of opening (*Figure 8B*), suggesting that anxiety negatively impacts successful helping in the presence of pro-social motivation. Rats were less active before door-opening compared to after door-opening (repeated measures ANOVA, main effect, p<0.001; *Figure 8C*). Pairwise comparisons found that this effect was due to significantly reduced activity in non-openers prior to door opening, which was not the case for openers (pairwise comparisons, p<0.05; *Figure 8C*). This difference may result from either

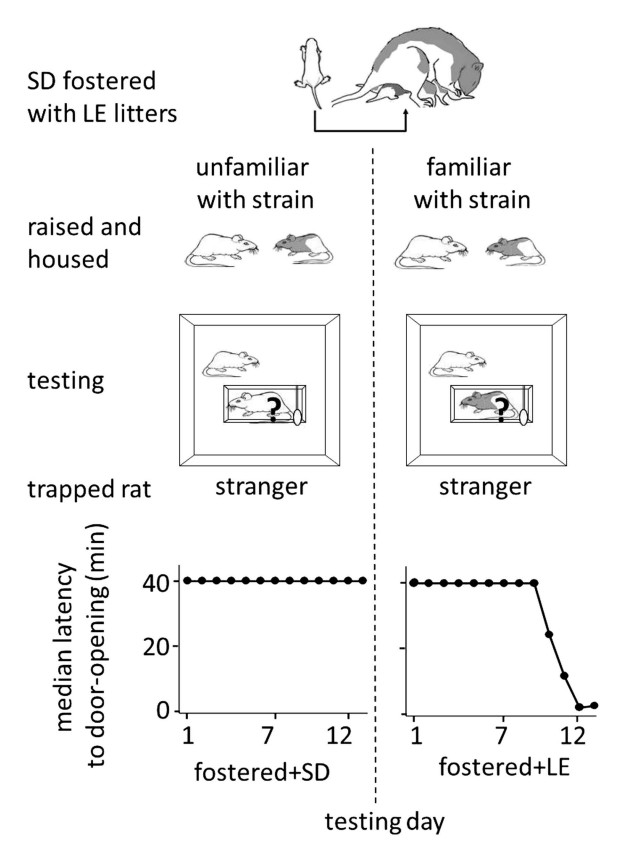

**Figure 4**. Strain familiarity, even to one's own strain, is required for the expression of helping behavior. Fostered SD rats were raised with LE rats from birth (top diagram) and were not exposed to or able to interact with other SD rats prior to testing. When fostered SD rats were adults, they were tested with trapped stranger rats ('?') of either the SD (left) or LE (right) strain. Fostered SD rats did not help trapped SD strangers. In contrast, fostered SD rats helped trapped LE strangers.

non-openers' greater anxiety or reduced interest in the trapped rat.

## Discussion

This study demonstrates that helping another rat, by releasing it from a restrainer, is flexibly applied to select others based on previous social experience. It is neither the individual identity nor the particular strain of the rat in need that motivates helping. Rather, it is the prior social experience of the free rat with any member of the trapped rat's strain that determines the target group for helping. Thus, rats help another rat from a given strain (same or different) only if they have previously lived with a member of that strain.

Rats pick up sensory cues from their cagemates that are retained and utilized to discriminate between strangers that share those cues and those that do not. This information then plays a key role in eliciting pro-social behavior, and could potentially be used for other behavioral decisions as well, when encountering an unfamiliar individual, effectively forming group categories. The experiments delineated above are not informative of the sensory modalities participating in the classification of strangers as similar or different to the cagemate. Previous research points to the importance of vision, olfaction and audition for affective communication between mice (*Langford et al., 2006*; *Jeon et al., 2010*). It is possible that helping behavior in rats similarly relies on visual, olfactory, auditory, and also tactile cues as the clear and perforated restrainer allows for full sensory communication between the free and trapped rats.

The present study effectively demonstrates that strain identity is meaningless without social experience during development. Fostered rats raised without social interactions with their own strain were not motivated to help strangers of their strain, compelling evidence against an innate bias for pro-social behavior towards one's own kind. This finding is congruent with studies showing that neither kinship nor perceived similarity is needed to motivate pro-social behavior in primates (*Batson et al., 2005*; *Horner et al., 2011*; *Baden et al., 2013*). In nature, social animals typically live with related others, and thus a pro-social bias favoring familiar others would also favor genetically similar others. Yet, relying on social experience rather than genetic similarity for guiding pro-social behavior has an added value in that it allows animals to flexibly adapt to different circumstances (*Dugatkin, 2002*). Assigning an affective meaning to a group category following social experience would be an efficient mechanism for appropriately extending pro-social behavior towards unknown individuals belonging to that same group.

While door-opening has an obvious pro-social outcome, a variety of motivations could contribute to this behavior. We have previously excluded the possibilities that door-opening depends on reward from either motor mastery or social contact (*Ben-Ami Bartal et al., 2011*). In addition it is unlikely that

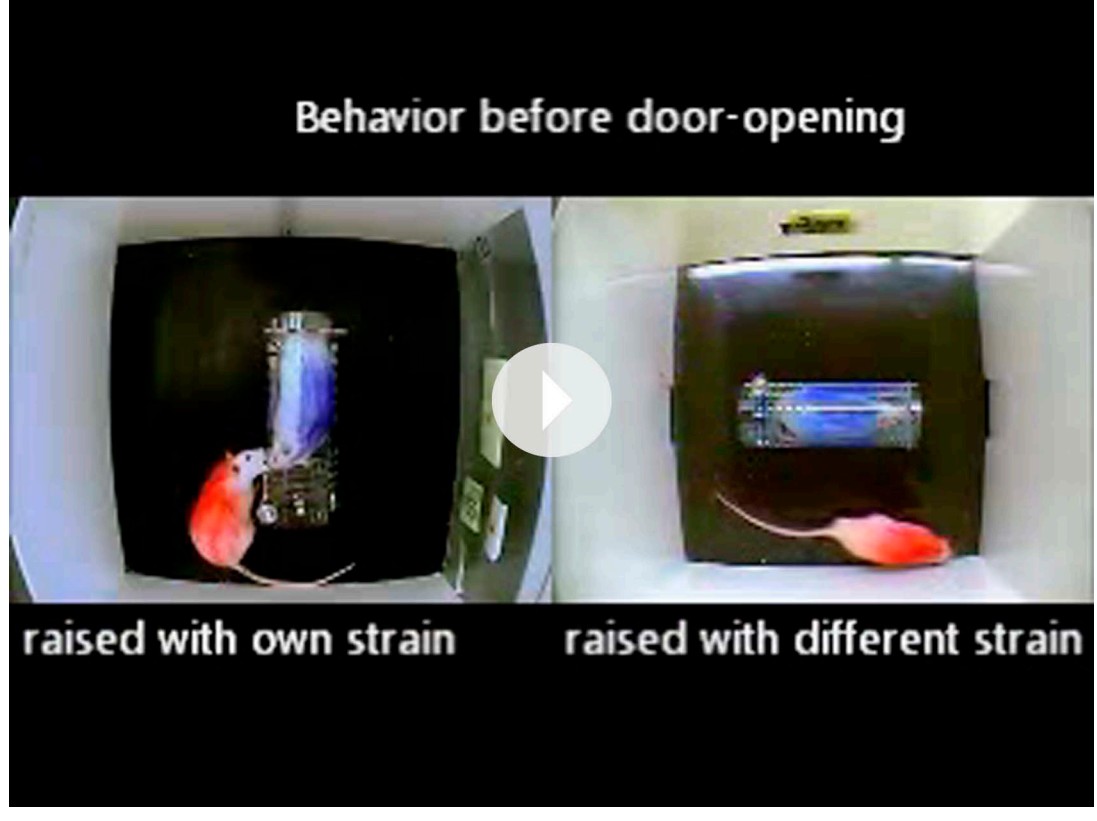

**Video 2**. Fostered rats do not help rats of their own strain. Normally rats show pro-social behavior toward rats of their own strain. However, SD rats raised exclusively with LE rats and in isolation from other SD rats did not show interest in or open for a trapped SD stranger.

rats were motivated to open the restrainer by aggressive or anti-social motivations (*Myer and White, 1965*) because conflict between the free and trapped rats was minimal for opener pairs. As non-openers tested with an unfamiliar strain were not less active before door-opening than openers tested with a familiar strain (*Figure 8C*), differences in general arousal cannot account for the observed results. Given the greater interest in the trapped rat shown by openers and the lack of aggressive interactions between opener pairs, the authors favor the interpretation that rats open the restrainer door in order to terminate the trapped rat's distress.

While we did not see any difference in helping behavior expressed toward strangers and cagemates, the underlying affective responses may have differed. Indeed, movement velocity was greater for rats tested with cagemates compared to those tested with strangers, indicative of different motivational states or magnitudes. Nonetheless, rats were sufficiently motivated to help strangers. This result is consistent with previous demonstrations of helping behavior expressed toward strangers in other animals, including humans (*Warneken and Tomasello, 2006*; *Batson et al., 2007*; *Loggia et al., 2008*; *Custance and Mayer, 2012*; *Tan and Hare, 2013*). While emotional contagion has been demonstrated for both individually familiar and unfamiliar mouse pairs (*Chen et al., 2009*; *Jeon et al., 2010*), it may be facilitated by individual familiarity (*Langford et al., 2006*; *Jeon et al., 2010*). Work in dogs and primates further supports a familiarity effect for emotional contagion (*Palagi et al., 2009*; *Campbell and de Waal, 2011*; *Silva et al., 2012*).

Empathy, the capacity to share and recognize the emotional states of another (*Decety, 2011*), often motivates approach and pro-social behavior and caring in humans (*Eisenberg and Miller, 1987*). The empathic response involves activation of common neural networks for processing one's own and another's distress (*Decety et al., 2012*). Some forms of empathy are primarily dependent on subcortical neural structures that are phylogenetically conserved across mammalian species (*Decety and Svetlova, 2012*). Emotional contagion, a fundamental component of empathy resulting

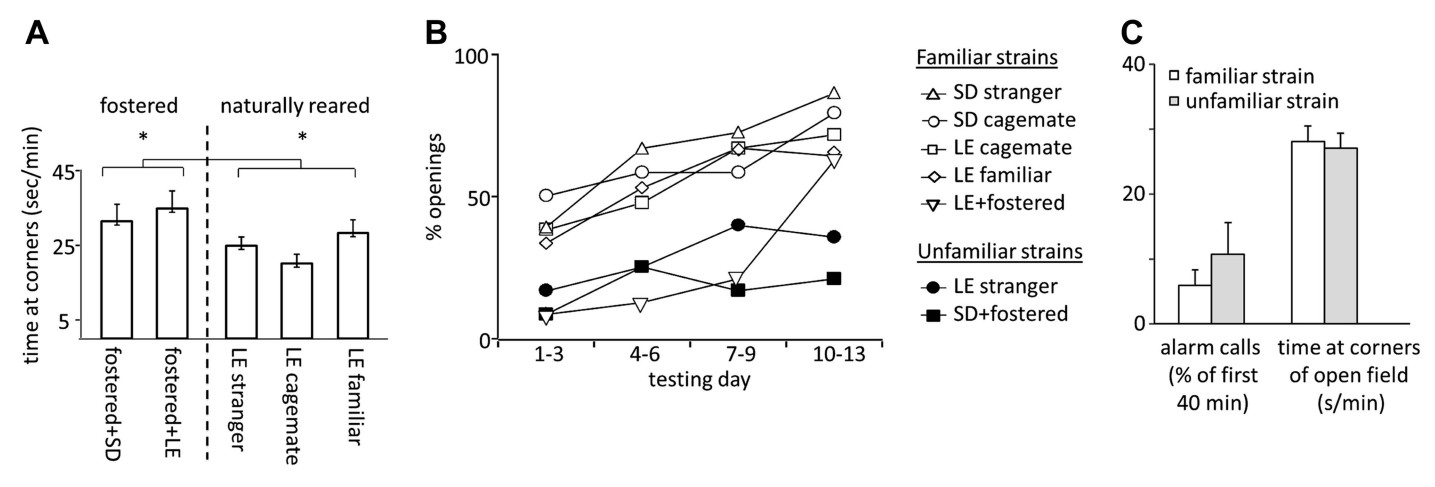

**Figure 5**. (**A**) Fostered rats (left of dashed line) spent more time at the arena corners during open field testing than naturally reared animals (right of dashed line). (**B**) The proportion of rats that opened increased across the testing sessions for rats tested with familiar strains (open markers) but not for those tested with unfamiliar strains (filled markers). (**C**) No differences between rats tested with trapped rats from familiar strains (white bars) and those tested with rats from unfamiliar strains (gray bars) were observed in the average number of alarm calls or the amount of time spent at corners during open field testing.

when the emotion of one individual evokes a matching emotional state in another individual (*Preston and de Waal, 2002*), has been demonstrated in rodents (*Chen et al., 2009*; *Langford et al., 2006*). Thus, rodents, and mammals in general, may share a mechanism for mobilizing pro-social motivation in response to the distress of another individual. We would argue that a rodent form of empathy, as defined above, is the main motivation for the helping behavior observed in our studies. Moreover, the absence of door-opening for rats of unfamiliar strains likely reflects a reduced empathic arousal of the free rats in these conditions.

Rats that are not familiar with a different strain do not act pro-socially towards others of that strain. The selectivity with which rats engage in helping behavior is further evidence that releasing a trapped rat is an intentional social behavior, congruent with an empathic drive to help some rats, but not others. Yet, social experience can modify this behavior. We conclude that through social interactions,

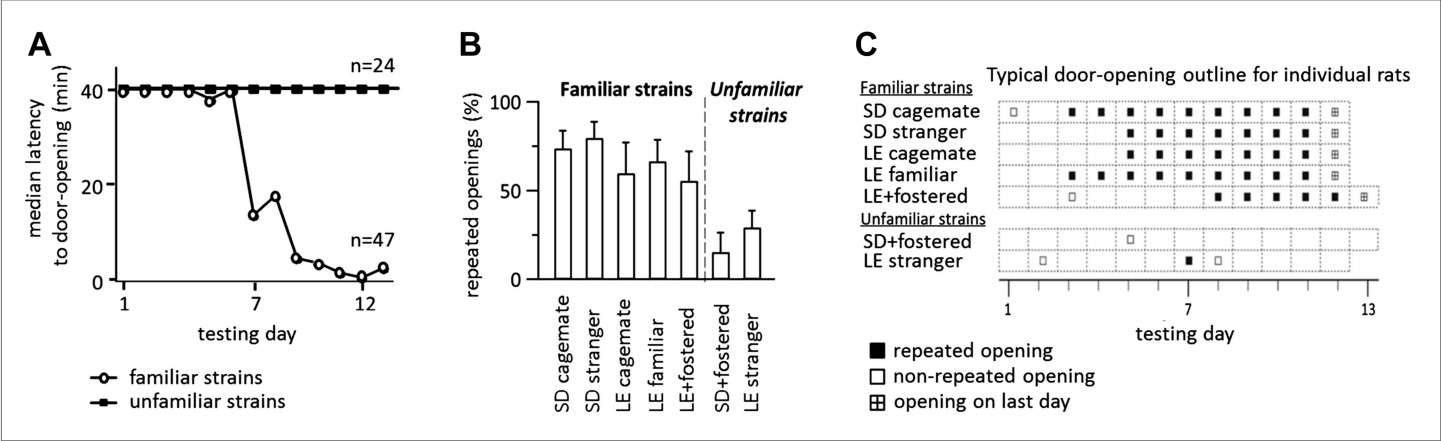

**Figure 6**. Rats experience helping rats of a familiar strain as rewarding. (**A**) The latency to door-opening decreased along the days of testing for rats tested with familiar strains (open circles) but not for those tested with unfamiliar strains (filled squares). (**B**) The proportion of openings that were followed by a repeated opening on the next day of testing was higher for trapped rats from familiar strains (left of dashed line) than from unfamiliar strains (right of dashed line). (**C**) Opening data from representative rats of each condition are illustrated. Fostered SD rats were tested for 13 days.

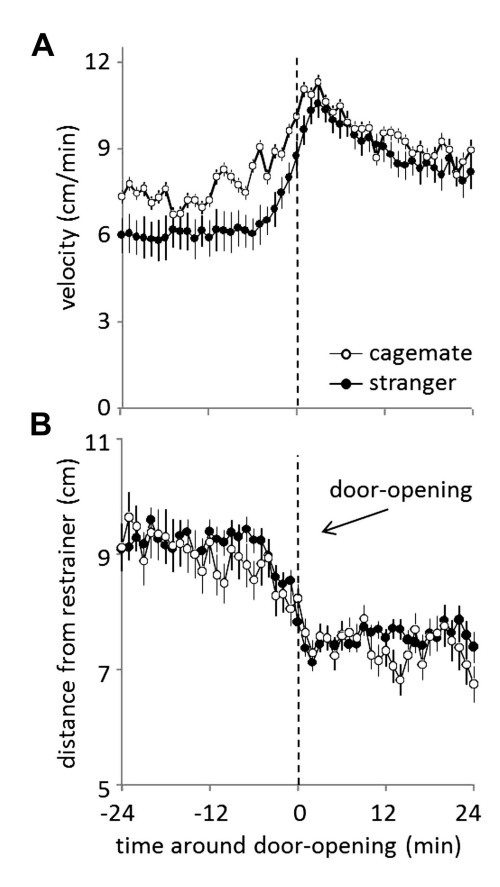

**Figure 7**. Prior to door-opening, rats are equally motivated to help, but show more activity for trapped cagemates than for strangers from a familiar strain. (**A**) Across conditions and testing days, rats tested with cagemates (white circles) were more active in the period before door-opening than were rats tested with strangers from a familiar strain (black circles). (**B**) The distance from the restrainer was not different for rats tested with cagemates (white circles) and strangers from a familiar strain (black circles).

rats form affective bonds that elicit empathy and motivate helping. This motivation to help is extended to strangers of familiar strains, showing that rats form groups based on social experience. As is the case for humans (*Xu et al., 2009*; *Mathur et al., 2010*), rats base the bias for pro-social behavior on group membership. And as for humans, a diverse social experience was effective in mitigating such bias (*Chiao et al., 2008*; *Elfenbein and Ambady, 2003*; *Madsen et al., 2007*; *Telzer et al., 2013*; *Zuo and Han, 2013*). Moreover, we have demonstrated that for rats, genetic similarity does not influence pro-social motivation. Rather, groups are socially defined. Whereas social bias is hard to overcome through cognitive effort (*Dovidio et al., 2002*; *Johnson et al., 2002*; *Amodio et al., 2004*; *Correll et al., 2007*), our results support the existence of at least one mechanism for altering group membership that does not require complex cognition.

## Materials and methods

### Subjects

Sprague-Dawley (SD) and Long-Evans (LE) male rats (Charles River, Portage, MI) were used for all studies. Rats were 8–11 weeks old at the start of the experiment. Rats were housed in pairs with *ad libitum* access to chow and water in a 12:12 light-dark cycle. Animals were allowed 2–3 weeks to acclimate to the housing environment. Stranger rats were always housed either in a separate room or on a separate rack than the free rats with which they were to be tested. Stranger rats were also handled separately, such that there was never any contact between strangers and free rats before the start of testing. The fostered SD rats were bred in-house from pregnant females purchased from outside (Charles River, Portage, MI; see more details below).

### Set up

Above every Plexiglas arena (50 × 50 cm, 32–60 cm high), a CCD color camera (KT&C Co, Seoul, Korea) was mounted. The cameras were connected to a video card (Geovision, Irvine, CA) in a dedicated PC.

Sound recordings of ultrasonic (15–70 kHz) vocalizations were recorded (Avisoft Bioacoustics, Berlin, Germany) through a single microphone in each testing room. Because recorded ultrasonic vocalizations could not be ascribed to individual rats and instead had to be assigned to each condition, all rats tested at one time in one room were always from the same condition.

### Restrainers

A Plexiglas rodent restrainer (25 by 8.75 by 7.5 cm, Harvard Apparatus, Holliston, MA) was placed in the center of the arena. The body of the restrainer had several small slits and the back end had a large slit, allowing for olfactory and tactile communication between rats. At the other end, a customized door had two panes that were attached with three screws, and a pole (5 cm) supporting two weights (25 g each). The weights were included in order to facilitate the door falling off to the side once the

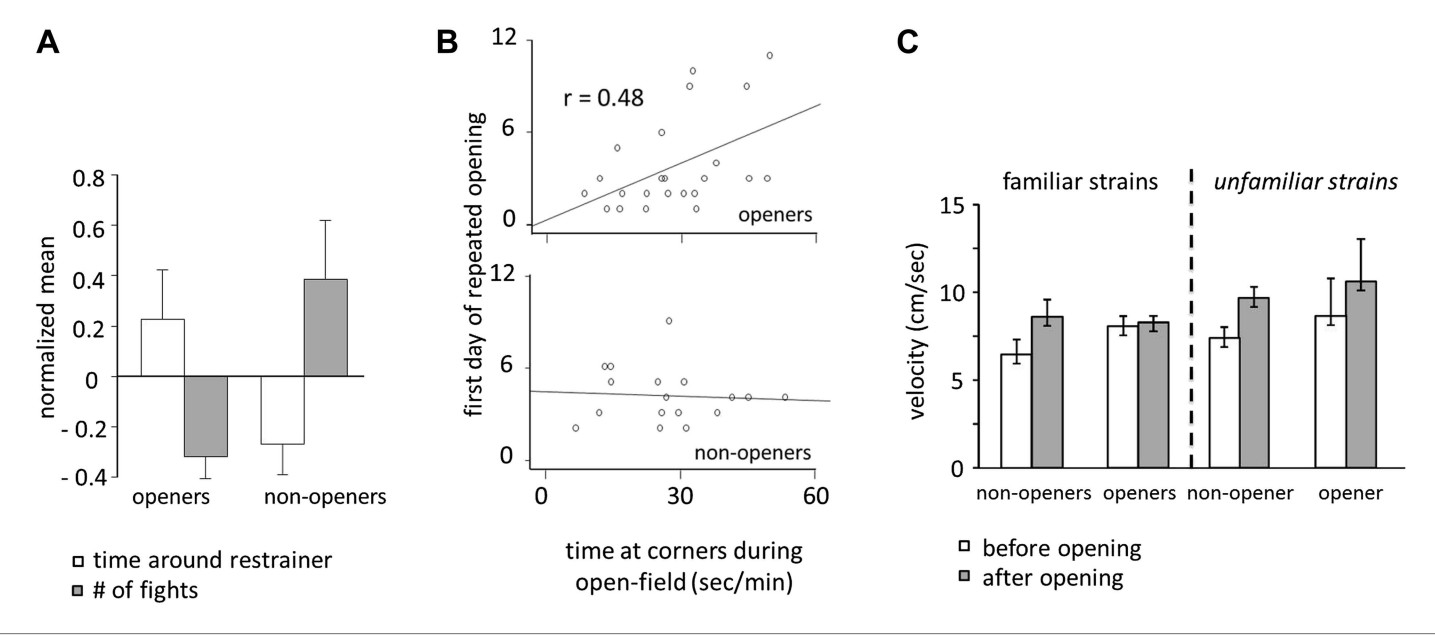

**Figure 8**. Openers showed an affiliative behavioral pattern. (**A**) Rats that became openers (left) spent more time around the closed restrainer (open bars) and fought less with the trapped rat (gray bars) than did non-openers (right). (**B**) The first day of repeated opening was correlated with time spent at the corners during open-field testing for openers (top) but not for non-openers (bottom). (**C**) Averaged across the 12 days of testing and regardless of strain familiarity, non-openers were less active when the restrainer was closed (white bars) than when it was open (gray bars).

free rat pushed on the door. The door was designed to be opened only from the outside. The free rat could open the door from the top, from the side, or by pushing up on the door with its snout.

## Handling
Animals were habituated to the experimenters (who were kept constant for each cohort of rats) and the arenas prior to being tested. On day 1, rats were transported to the testing room and left undisturbed in their home cages. On day 2, rats were briefly handled and tested for 'time-out' (see below). Starting with the second day of habituation, rats were weighed three times each week for the duration of the experiment; no animal lost weight during the experiment. On days 3–5, rats were tested for time-out, marked, and handled for 5 min by each experimenter. Rats were then placed with their cagemate in the testing arenas for 30 min. All free rats and all cagemates were always placed in the same arena for habituation and testing. Rats that were used as strangers were placed in the same arena for habituation but in different arenas during testing. Habituation of free rats and strangers never occurred at the same time. After each habituation session, rats were returned to their home cages and to the housing room. All sessions were run during the rats' light cycle between 0800 and 1730. Order of testing was counterbalanced between sessions to control for effects of time of day on behavior.

## Time-out measurements
Time-out was measured as the latency from opening the homecage lid halfway to the time that the rat approached the front edge of the cage, reared up, and placed its paws on the ledge. This measurement was recorded 3–5 times for each rat in every cage during habituation.

## Open field testing
On the day following completion of habituation, rats were placed individually in an arena for 30 min and their activity recorded. Note that the arenas were the same as were used during habituation but that open field testing was the first time each rat had been in the arena alone.

## Testing procedures
At the start of testing, the trapped rat was placed in the restrainer, the door closed, and the restrainer placed in the center of the arena. The free rat was then placed in the arena. If the free rat did not open

the restrainer door within 40 min, the investigator opened the restrainer door halfway, to a 45° angle, greatly facilitating door-opening by either rat. Regardless of whether the door was opened before or after the 40 min mark, both rats always remained in the arena for the full hour-long session. After each session, the arena and restrainer were washed with 1% acetic acid followed by surface cleaner. Rats were tested once daily for 12 days.

If a trapped rat succeeded in opening the door from inside the restrainer (~30% of rats), the trapped rat was placed immediately back in the restrainer, and a Plexiglas blocker was inserted, preventing his access to the door. If the free rat subsequently opened the door, the blocker was removed, allowing the trapped rat to exit the restrainer. The blocker was then used for that trapped rat on all following test days.

## Protocols

The free rat was a male SD rat in all conditions. The strain of the male trapped rat was either SD (SD cagemate, SD stranger, fostered+SD) or LE (LE cagemate, LE stranger, LE familiar, fostered+LE) as described further below. All rats used in all conditions were pair-housed.

### SD cagemate and SD stranger conditions

Rats for the SD cagemate conditions were housed together; the rat with the shorter time-out latency was used as the free rat. The time-out latency was not used to determine the free rat in any other condition. Stranger rats, destined to serve as trapped rats, were housed and habituated separately from the free rats. Following habituation, free rats were tested with trapped cagemates (n = 8) or strangers (n = 12). In the stranger condition, each free rat was exposed to a different stranger on every day of testing.

### LE cagemate and LE stranger conditions

On the day of arrival from the vendor, each SD rat in the LE cagemate condition (n = 8) was housed with an LE rat of comparable size. Rats were observed for an hour after this pairing and no vigorous fighting or injuries occurred. In most pairs, the rats were asleep in a huddle before the hour was up. Rats in the LE cagemate condition remained pair-housed until the end of testing. One LE rat died and the remaining SD rat was excluded from the experiment. Hence, there were only seven rat pairs tested in the LE cagemate condition.

In the LE stranger condition, SD rats (n = 16) were pair-housed with each other and separately from the LE trapped strangers. Habituation to the testing environment occurred separately so that free SD rats were never exposed to LE rats prior to testing. Half of the LE strangers were pair-housed with other LE rats, and half were pair-housed with an SD rat. There was no difference in helping behavior between the two treatments and therefore, the data were pooled.

### LE familiar condition

Each free rat (SD males, n = 12) was pair-housed with an LE rat for 2 weeks. On day 14, SD rats were re-housed with each other and allowed to readjust for 1 week prior to habituation. Rats were then tested with trapped LE strangers as described above. Two cages of SD rats (four rats) arrived 4 days late and were housed with LE rats for 10 days instead of 14. Two of those rats became openers and the other two were non-openers.

### Fostered SD conditions

Multiparous pregnant LE (n = 16) and SD (n = 7) dams were purchased (Charles River, Portage, MI). Two SD pups were placed into each LE litter on either P0 or P1 (average 0.3 ± 0.1). At this time, which corresponded to P0 of the LE litter (0.3 ± 0.2), LE litters were culled to seven LE pups except for one litter which had only three LE pups. When the SD pups were on average 6.4 ± 0.2 days old, one SD pup was removed, leaving a single male SD pup in the LE litter. Thus, most SD male pups (14/16) were exposed to one SD littermate for no more than 7 days (days when the pups' eyes were closed). However, in two cases, the SD pup in the LE litter died at a later date (P11, P12). In these cases, an SD male who had been removed from an LE litter 5 or 7 days before was then re-added to the LE litter. In sum, the average exposure to one or more SD pups was 7.2 ± 0.4 days. The pup with the maximal exposure to at least one SD pup (11 days) was transferred when his eyes were open; this rat became the sole opener in the fostered+SD condition.

Pups in the fostered SD conditions were weaned at 28 days of age and pair housed with a male LE littermate. At 2 months of age, the SD rats were tested with either SD strangers (n = 8) or LE strangers (n = 8). Both SD and LE strangers were purchased as adults and allowed to acclimate for 2 weeks prior to habituation and testing.

## Video analysis

Ethovision tracking software (Noldus Information Technology, Inc. Leesburg, VA) was used to track the rats' movements in the arena. To enable tracking both rats' movements, free rats were colored red and trapped rats colored blue. The rat's location was converted into x, y coordinates denoting the rat's location at each frame at a rate of 7.5 FPS. These data were then used to calculate movement velocity and location in the arena (time around the restrainer, time at arena corners).

## Behavioral coding

Freeware (Jaywatcher V1.0) was used to manually code the rats' interactions for 15 min following door-opening. If the door was opened less than 15 min before the end of session, only the remaining time in the arena was coded. Data were not analyzed for five rats (four SD stranger, one LE cagemate) due to technical problems. Coded behaviors included anogenital sniffing, pinning, wrestling, and boxing. Coding was performed by four judges. There was an 84% agreement between judges. Boxing was the behavior that was selected to represent fighting, and analyzed. One rat (a non-opener in the SD cagemate condition) boxed on 95 occasions, a total that was more than five standard deviations over the mean for all rats tested. Additionally, this rat boxed on 10 of the 12 testing days, more than two standard deviations over the mean. This rat was therefore considered an outlier and was excluded from the boxing analysis.

## Audio analysis

Vocalizations during the first 40 minutes of the session were analyzed for days 1, 3, and 9 of testing using Avisoft SASLab Pro (Berlin, Germany). Audio data was not available for rats in the SD cagemate and SD stranger conditions (data were lost due to disk failure). Thus, audio data came from rats in the LE cagemate, LE stranger, LE familiar, and both fostered conditions.

## Mothering style analysis

The mothering behavior of all dams was recorded at 5 min intervals for three hours (an hour each, starting at 0900, 1200, and 1600) between P1 and P10 (methods adapted from *Champagne et al., 2003*). The nursing behaviors recorded were (1) arched back nursing; (2) blanket nursing; or (3) passive nursing. In addition, we recorded if the dam was either (4) not in contact with pups; or (5) licking or grooming a pup. The proportion of time bins within each of the five categories was calculated for each dam on each day. The proportions of each behavior recorded were not different between rats in fostered+LE and fostered+SD conditions (1-way repeated measures ANOVAs, p>0.11).

## Door-opening latencies

Time to door-opening was calculated as the minute when the restrainer door was opened minus the start time. For rats that never opened, a cutoff time of 40 min was assigned.

## Definition of openers

Rats that opened the restrainer on two sequential days, and did so at least three times were termed 'openers'. Thus, a rat that opened the restrainer 4 days in a row was considered an opener. In all cases except one, once this criterion was met, the rat continued to open the restrainer until the end of the experiment. In the one exception, the rat opened the restrainer on days 4–8 but not on days 9–11. This rat (in the LE familiar group) was considered a non-opener. Three rats (SD cagemate, LE cagemate, and fostered+LE conditions) opened on the final 3 days of testing and did so at decreasing latencies. These rats were considered openers.

## Definition of first day of learned opening

The day on which opener rats learned to open the restrainer was defined as the first opening that was repeated the next day.

## Statistical analysis

Door-opening latencies, velocity, time in arena corners, and time around the restrainer were averaged per rat across all sessions and all days. ANOVA and two-tailed Student's *t* tests were used to determine differences between groups. Fischer PLSD was used for all post-hoc analyses. A chi-square analysis was used to compare the proportions of openers and non-openers. In all cases, α <0.05 was used as criterion for significance. Door-opening latencies are displayed using the median since door-opening was not a normally distributed variable. Statistical comparisons were conducted using SPSS (PASW 18).

## Acknowledgements

The assistance of Nora Molasky, Tony Logli, Michelle Yun, Kirill Karlin, Joshua Saucedo, Calvin Krogh, and Victoria Huang is gratefully acknowledged.

## Additional information

### Competing interests

PM: Reviewing editor, *eLife*. The other authors declare that no competing interests exist.

### Funding

| Funder | Grant reference number | Author |
|---|---|---|
| National Institutes of Health | DA022978, DA022429 | Peggy Mason |

The funder had no role in study design, data collection and interpretation, or the decision to submit the work for publication.

### Author contributions

IB-AB, DAR, PM, Conception and design, Acquisition of data, Analysis and interpretation of data, Drafting or revising the article; MSBS, Acquisition of data, Analysis and interpretation of data; JD, Contributed critically to the conceptual and theoretical framework for this work, Drafting or revising the article

### Ethics

Animal experimentation: All animal husbandry and all experiments were performed in accordance with the National Institute of Health guidelines and approved by the Institutional Animal Care and Use Committee of the University of Chicago (Animal Care and Use Protocol #71967). Every effort was made to minimize the number of rats used.

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
