## [Decision Letter]

Thank you for sending your work entitled “Kindness to strangers depends on knowing their kind: Pro-social behavior in rats is modulated by social experience” for consideration at *eLife*. Your article has been favorably evaluated by a Senior editor, a Reviewing editor, and 3 reviewers.

The Reviewing editor and the other reviewers discussed their comments before we reached this decision, and the Reviewing editor has assembled the following comments to help you prepare a revised submission.

We have received three reviews of your submitted manuscript, all of which found the evidence you presented that rats preferentially help other rats from the same group plausible but not necessarily the only possible interpretation. There are several important concerns that need to be addressed based on the reviewer comments. A consistent, central theme of the reviews that needs to be addressed is the confounding of description with explanation as outlined in the first conceptual issue. The conceptual and practical issues are enumerated below.

Conceptual issues:

1) The authors state in the Introduction that the “release of cage mates … provides evidence for empathically motivated helping…” In subsequent prose, the results presented in the present paper are described in terms of “empathy” and “helping”, confounding description with explanation. The empirical facts are that rats, under some situations, will release another rat from an enclosure, however, that this is described as an act of ‘prosocial empathy and with the aim of helping the other animal’ which constitutes an explanatory hypothesis. The evidence in neither the previous paper nor this one convinces me that ‘empathy and helping’ are the most likely explanations. A lot more work is needed to tease out simpler explanations. For example, rats will work for the opportunity to engage another rat in a fight ([43], Animal Behaviour, 13, 430-433) or in affiliative play (Humphreys & Einon, 1981, Anim Behav, 29, 259-270), and in unfamiliar settings adult males will engage in a form of play that incorporates elements of aggression (Smith et al. 1999, Aggress Behav, 25, 141-152). Thus, a mixture of prosocial and anti-social motivations may be involved in the paradigm used in this paper. I am only half convinced that the release of a conspecific is unambiguously due to prosocial motivation. Even less convincing is that this releasing behavior arises from empathy. There are a large number of added cognitive processes that need to be in place for this to be the case. If the trapped rat is truly in discomfort in the enclosure, it is likely to be emitting many 22kHz calls (as recorded in this paper as alarm calls). While rats are attracted to 50kHz calls (Wohr & Schwarting, 2007, PLOS ONE, 2, e1365), hearing 22-kHz induces freezing and avoidance (Brudzynski & Chiu, 1995, Physiol Behav, 57, 1039-1044), so what is it about the “trapped rat” that is attracting the performer? Empathy and helping may account for the performer’s coming to the rescue, but to do so, it would need to somehow suppress its own defensive reaction to the alarm calls, as is implied by the authors’ use of the words empathy and helping. But surely, this is just one of the many possible explanations for the releasing behavior. The authors need to revise the manuscript to reflect a more nuanced explication of the data in descriptive terms that are more neutral. Moreover, the concerns of alternative explanations described above need to be addressed.

2) An alternative explanation for the results that “the target rats are biased to release those rats with which they are familiar with not because of any empathic feelings towards the trapped rat, but that they are compelled to release these rats so that they can play with them, which then serves as a social reward.” The videos provided by the authors show playful interactions, including pouncing, pinning, and other measures of play in the rat. The authors should address this possibility and describe how they view this alternative explanation.

3) Citations that seem relevant are missing, including those reporting co-residence duration and maternal perinatal association known to be major cues for human helping behavior and moral judgment (see Lieberman et al., “The Architecture of Human Kin Detection”, 2007; Fessler and Navarrete “Third-party attitudes toward sibling incest”, 2004; Lieberman and Lobel, “Kinship on the Kibbutz: co-residence duration predicts altruism, personal sexual aversions and moral attitudes among communally reared peers”, 2012 for good examples ). Similar effects have been observed in non-human species as well, such as long-tailed tits that tend to help other long-tailed tits (by aiding in the rearing of non-offspring) they have associated with during the nesting phase (see West et al. “Evolutionary Explanations for Cooperation”, 2007 for a review). Specific social cues (especially those perceived during rearing) can be more important than genetic similarity for predicting helping behavior. Clarify how the reported results relate to this literature and what is novel in this context.

4) The authors should acknowledge that previous reports suggest rats would help their cagemates more than strangers, which is contrary to what was shown here.

5) Some literature was at least slightly misrepresented, for example, though the Van Bavel et al. 2008 study did show some interesting effects of social experience on hemodynamic responses to ingroup/outgroup faces, it did not claim to show anything about empathy or pro-social behavior. These data need to be correctly represented.

Practical issues:

1) The authors use “stock” throughout most of the paper but then use the term “strain” in Figures 4 and 5. Please make this consistent by using strain that is the more generally understood term.

2) Some terms are used loosely, including describing a rat approaching the front edge of the cage as “bold” but it could also be curious. Since “boldness” was used to choose rats to be “free” this needs to be more carefully defined. Also, boxing is used a measure of fighting yet is described among other behaviors that seem somewhat playful. These should be distinguished. Similarly, clear definitions of intentional, empathy, altruism and pro-social need to be provided.

3) Explain why the audio recordings from only testing days 1, 3, and 9 were analyzed.

4) The 1st panel of Figure 6 is very confusing, given that the “opener” rats would have opened the door at different times – is the claim that each rat’s locomotion decreased immediately after they opened the restrainer, or rather, that it decreased mainly during the last twenty minutes of the session?

5) Figure 3 is confusing. Is the fostered rat that will be tested not depicted in each illustrated litter? What is happening with the first 2 litters that switch from mostly LE to all SD and then back to LE in panel B? Please clarify.

[Editors' note: further clarifications were requested prior to acceptance, as described below.]

Thank you for resubmitting your work entitled “Pro-social behavior in rats depends on social experience not genetic relatedness” for further consideration at *eLife*. Your revised article has been evaluated by a Senior editor, a member of the Board of Reviewing Editors, and the three original reviewers.

Your revised manuscript has been considerably improved but still has several issues needing attention. Overall, the reviewers continue to feel that the data do not support an interpretation of empathy and therefore some of the writing suggesting empathetic motivation remains and needs to be eliminated. Specific comments are summarized here:

1) General arousal seems to be a very reasonable alternative to the pro-social interpretation of your results. This explanation needs to be presented as an alternative and discussed.

2) The present study doesn’t really contrast an ‘experience’ versus a ‘genetic’ hypothesis. What is really interesting is that a rat will treat strangers that look like the rats with whom they were reared as if they were familiars, or least ones that they will preferentially release. The rats single out some cue(s) about their cage mates that they can then apply to strangers and discriminate among them as those that share the cues with cage mates and those that do not. The authors should discuss this and consider what cues might be being used.

3) A related concern is that there is no explanation of the observation that the door was opened for cagemates as often as for strangers. Every paper published thus far across species (including mice, rats, monkeys, and humans) shows increased “helping” or “cooperation” or “empathy” for cagemates vs non-cagemates, and there is a lot of theory justifying why this should be the case. This suggests that there is something about the paradigm that blocks social cues relevant for individual recognition. If you want to suggest something so out of line with previous results you either need to rule out alternative explanations with more experiments or more honestly acknowledge/discuss why these results differ, especially when the claim is so central to the significance of the paper.

4) The authors dismiss the concerns raised that the releasing may be due to social reward because their previous findings found that releasing can occur in the absence of subsequent contact. For personally acquainted rats, immediate reward may not be needed because of the past association made between access to an individual and rewarding experiences. Similarly, by showing that unfamiliar rats with some cue identifier marking them as similar to cage mates are preferentially released, it may be simply a two-step association process. Release of a rat with a similar cue may occur because of a past association with social reward. Then, in the present experimental paradigm, the releasing rat may actually obtain the social reward, reinforcing the releasing behavior. That is, in this case the social reward may be important to release a rat that is not familiar, but resembles cage mates. The present findings may be accounted by simple association mechanisms, not requiring higher order processes involving empathy. You should discuss the role of association learning in interpreting your results.

5) “…the most parsimonious explanation is that rats open the restrainer door in order to terminate the trapped rat’s distress”. This sentence should be replaced with something like: “A possible interpretation of the data is that rats open the restrainer door in order to terminate the trapped rat’s distress or in order to terminate their own vicarious arousal.”

6) “The selectivity with which rats engage in helping behavior is further evidence that releasing a trapped rat is an intentional social behavior, congruent with an empathic drive to help some rats, but not others… We conclude that rats form affective bonds through social interactions that motivate empathy and helping. The motivation to help is extended to strangers of familiar strains…” This is an example of how the current version of the paper still concludes that the observed behavior represents rats consciously and intentionally helping others (altruism). As all the reviewers agreed, this conclusion is not warranted.

7) Please cite the [52] paper on how social experience modulates generalized reciprocity in rats, and modify the relevant language in the abstract and discussion to give proper credit to this prior study. The generalized reciprocity mechanism discussed by [52] may be a useful explanation for the social experience-modulated door-opening behavior described here as well.

---

## [Author Response]

[Editors’ note: the author responses to the first round of peer review follow.]

*Conceptual issues*:

*1) The authors state in the Introduction that the “release of cage mates … provides evidence for empathically motivated helping…” In subsequent prose, the results presented in the present paper are described in terms of “empathy” and “helping”, confounding description with explanation. The empirical facts are that rats, under some situations, will release another rat from an enclosure, however, that this is described as an act of ‘prosocial empathy and with the aim of helping the other animal’ which constitutes an explanatory hypothesis. The evidence in neither the previous paper nor this one convinces me that ‘empathy and helping’ are the most likely explanations*.

The reviewers were concerned that while we describe a pro-social act that is directed in a socially selective way, we then explain that behavior as resulting from empathic motivation. We completely agree that the data presented here, on their own, do not prove that empathy motivates the rats’ helping behavior. Further, attributing a specific motivation to door-opening is unnecessary for our central conclusion that rats show an experience-based rather than a genetic- based social selectivity for helping. Therefore, we have carefully revised the text to focus on the social modulation of pro-social behavior, rather than on the underlying affective motivation for this behavior.

In the Discussion we briefly review potential motivations for the observed behavior and argue that the balance of available evidence suggests that a rodent form of empathy is the most likely motivation for door-opening.

*A lot more work is needed to tease out simpler explanations. For example, rats will work for the opportunity to engage another rat in a fight (*[43]*, Animal Behaviour, 13, 430-433) or in affiliative play (Humphreys & Einon, 1981, Anim Behav, 29, 259-270), and in unfamiliar settings adult males will engage in a form of play that incorporates elements of aggression (Smith et al. 1999, Aggress Behav, 25, 141-152). Thus, a mixture of prosocial and anti-social motivations may be involved in the paradigm used in this paper*.

This comment raises a very important conceptual point that we now discuss explicitly in the manuscript.

While it is true that a variety of motivations could contribute to door-opening, it is unlikely that rats were motivated to open the restrainer by aggressive or anti- social motivations. Our results show that conflict between the free and trapped rat was minimal between opener pairs. Moreover, openers exhibited a behavioral pattern indicative of a pro-social motivation, such as constantly circling the restrainer (indexed by proximity to the restrainer), creating tactile contact with the trapped rat, digging and biting the restrainer, and making multiple attempts to release the trapped rat. The free rat’s behavior was often reciprocated by the movements of the trapped rat, as the latter changed his position in the restrainer to face the free rat. The non-openers, on the other hand, spent less time around the restrainer and were more aggressive when the trapped rat exited the restrainer following half-way door-opening. Thus, we find it unlikely that free rats released the trapped rat in order to engage in a fight.

The rats might have released the trapped rat in order to obtain social contact such as playing. However, we have previously established that social reward is not required for helping. Thus, at least some other source of motivation must be present. We would argue that this other motivation is to terminate the distress of the other rat, a rodent form of empathy. Yet, as mentioned above, the evidence provided by this study alone does not provide stand-alone evidence for this claim, and we have revised the manuscript to reflect this.

*I am only half convinced that the release of a conspecific is unambiguously due to prosocial motivation. Even less convincing is that this releasing behavior arises from empathy. There are a large number of added cognitive processes that need to be in place for this to be the case*.

We completely agree that rodents do not possess the complex empathic abilities manifested in humans, which include perspective taking and other functions of abstract cognition. However, the existence of a simple, primitive form of empathy in many mammals is by now widely accepted (51; 35; 52; 10; 31; 48; 18; 42; 47; 49). It should be noted that fundamental empathic responses, such as emotional contagion and emotional resonance, are believed to rely on subcortical circuits rather than complex cognition (18).

*If the trapped rat is truly in discomfort in the enclosure, it is likely to be emitting many 22kHz calls (as recorded in this paper as alarm calls). While rats are attracted to 50kHz calls (Wohr & Schwarting, 2007, PLOS ONE, 2, e1365), hearing 22-kHz induces freezing and avoidance (Brudzynski & Chiu, 1995, Physiol Behav, 57, 1039-1044), so what is it about the “trapped rat” that is attracting the performer? Empathy and helping may account for the performer’s coming to the rescue, but to do so, it would need to somehow suppress its own defensive reaction to the alarm calls, as is implied by the authors’ use of the words empathy and helping*.

The reviewers appear concerned that the trapped rat is simultaneously not in discomfort and also frozen with distress. In fact, the truth appears to lie somewhere in the middle. The “restraint” in this study does not cause physical pain or extreme distress. The rats move and turn around within the restrainer. And they are never trapped for more than 40 mins. Nonetheless, the rats’ attempts to escape (30% of the trapped rats managed to open the door from the inside) make it clear that they find being in the restrainer aversive. Unpublished experiments show that all trapped rats given the opportunity to get themselves out (the door was reversed) of the restrainer do indeed take this opportunity.

It is our impression that the moderate stress level produced by the restraint prevents the extreme reaction of freezing immobility. This impression is supported by the low rates of ultrasonic vocalizations in the 22 kHz range (<6 min of alarm calls per 40 min session per 8-12 trapped rats) recorded.

The question of what feature of the trapped rat triggers the free rat’s actions is an interesting one. Research in mice suggests that emotional contagion is dependent on vision (35). We do not at present have any evidence regarding the modality by which the trapped rat communicates his distress.

*But surely, this is just one of the many possible explanations for the releasing behavior. The authors need to revise the manuscript to reflect a more nuanced explication of the data in descriptive terms that are more neutral. Moreover, the concerns of alternative explanations described above need to be addressed*.

As discussed above, we have carefully revised the manuscript to speak of pro- social behavior without committing to the motivation involved. Additionally, we have added a paragraph (see paragraph beginning, “While door-opening has an obvious pro-social outcome, a variety of motivations could contribute to this behavior…”) to the Discussion that addresses potential alternative explanations.

*2) An alternative explanation for the results that “the target rats are biased to release those rats with which they are familiar with not because of any empathic feelings towards the trapped rat, but that they are compelled to release these rats so that they can play with them, which then serves as a social reward.” The videos provided by the authors show playful interactions, including pouncing, pinning, and other measures of play in the rat. The authors should address this possibility and describe how they view this alternative explanation*.

Social reward is a fundamental underpinning of all social behavior. We agree that at root, behavior involving more than one individual is fundamentally motivated by social reward. However, as mentioned above, our previous results show that pro-social behavior occurs even in the absence of the possibility of social play (Ben-Ami Bartal et al., 2011). This is mentioned in the revised manuscript.

*3) Citations that seem relevant are missing, including those reporting co-residence duration and maternal perinatal association known to be major cues for human helping behavior and moral judgment (see Lieberman et al., “The Architecture of Human Kin Detection”, 2007; Fessler and Navarrete “Third-party attitudes toward sibling incest”, 2004; Lieberman and Lobel, “Kinship on the Kibbutz: co-residence duration predicts altruism, personal sexual aversions and moral attitudes among communally reared peers”, 2012 for good examples ). Similar effects have been observed in non-human species as well, such as long-tailed tits that tend to help other long-tailed tits (by aiding in the rearing of non-offspring) they have associated with during the nesting phase (see West et al. “Evolutionary Explanations for Cooperation”, 2007 for a review). Specific social cues (especially those perceived during rearing) can be more important than genetic similarity for predicting helping behavior. Clarify how the reported results relate to this literature and what is novel in this context*.

The kibbutz studies show that people expand their pro-social behavior toward individuals that they were reared with and that this occurs even for non-kin. Our findings are different in two very important ways from what was found in the kibbutz studies:

• The kibbutz studies show that living with another during childhood increases the likelihood of altruism expressed towards that one individual. In contrast, we show that social interaction (during either rearing and adulthood or adulthood only) with members of a strain leads rats to help strangers of that strain. The generalization from one individual to others of that individual’s strain is novel.

• The kibbutz studies show that early experience can influence pro-social motivation but do not rule out the possibility that genetic relatedness could influence helping behavior. We show that for rats, the genetic identity of the rat in need does not in any way predicate helping. The only determinant of helping is social experience, i.e., creating a social bond with a type of rat. The demonstration that genetic relatedness alone is not capable of producing pro-social motivation is novel.

There are numerous additional differences between our study and the kibbutz studies (timing of co-habitation, highly socialistic ideology, and culture in kibbutzim, and so on). Therefore, although we share the reviewers’ enthusiasm for these very interesting papers, we do not believe they are directly relevant and do not cite them.

Cooperation between tits is now cited in the Introduction (27).

*4) The authors should acknowledge that previous reports suggest rats would help their cagemates more than strangers, which is contrary to what was shown here*.

As far as helping behavior in rodents, we are not aware of studies that directly compared pro-social behavior toward cagemates and strangers (e.g., [12]; [52]). However, there is a robust literature that emotional contagion in both rats and mice is greater for familiar conspecifics than for unfamiliar ones. The revised manuscript includes a sentence referencing this issue: “…it is important to note that although we did not observe differences in helping towards strangers and cagemates of a familiar strain, the underlying affective response to strangers and cagemates may differ, as is suggested by research in apes (8) and mice (35).”

*5) Some literature was at least slightly misrepresented, for example, though the Van Bavel et al. 2008 study did show some interesting effects of social experience on hemodynamic responses to ingroup/outgroup faces, it did not claim to show anything about empathy or pro-social behavior. These data need to be correctly represented*.

We thank the reviewers for pointing this out, and we have corrected the manuscript with regards to the Van Bavel paper. We have also double-checked our representations of other results.

*Practical issues*:

*1) The authors use “stock” throughout most of the paper but then use the term “strain” in*
Figures 4 and 5*. Please make this consistent by using strain that is the more generally understood term*.

Done.

*2) Some terms are used loosely, including describing a rat approaching the front edge of the cage as “bold” but it could also be curious. Since “boldness” was used to choose rats to be “free” this needs to be more carefully defined*.

We agree completely that the meaning of the latency to approach the front of the cage is ambiguous. We now use the neutral term “time-out” to refer to this latency.

To be clear, the time-out measure was used in our previous study (Ben-Ami Bartal et al., 2011) but was only used to choose the free rat in the SD cagemate condition in the present study. In the other 6 conditions of the current study, the chosen rat was either the only SD rat in the cage (LE cagemate, 2 fostered conditions) or both rats in the cage were tested (SD stranger, LE stranger, LE familiar).

Note that the opening behavior of rats in SD stranger, LE cagemate, fostered+SD, and LE familiar conditions was not different from that of the only condition in which time-out was used to choose the free rat. Therefore, a short time-out is clearly not a requirement for motivating door-opening.

*Also, boxing is used a measure of fighting yet is described among other behaviors that seem somewhat playful. These should be distinguished*.

Most component movements in playing and fighting are the same (Meaney and Stewart, 1981). For example, pinning and wrestling can occur in bouts that appear to be “playful” and result in no injury as well as in interactions that escalate into boxing, biting and injury. We chose to analyze boxing which: 1) tended to appear only after intense, escalating interactions; 2) was often preceded by a lateral threat; 3) was always present in interactions that resulted in bites.

Similarly, clear definitions of intentional, empathy, altruism and pro-social need to be provided.

We now provide definitions of empathy and pro-social behavior:

• pro-social behavior comprises actions that improve the well-being of others

• empathy is defined as recognizing and sharing the emotional state of another

We do not use the terms altruism or intentional and therefore did not define them.

*3) Explain why the audio recordings from only testing days 1, 3, and 9 were analyzed*.

Rats were tested in a single room and not in soundproof chambers. A single microphone was used to collect the acoustic output of the group. Therefore, the audio recordings provide information about condition but not about individual rats. Because the audio recordings cannot be assigned to individual rats or even to one pair, examining the calls emitted before a pair’s first door-opening, for example, would not be informative. Since different pairs open on different days, there is no one day that reliably captures, for example, the day before first opening.

Given that only condition information can be gleaned from these recordings, we chose to sample calls that occurred early on (days 1 and 3) as well as later in testing (day 9). Sampling more days would be unlikely to provide novel information or insight and would likely overestimate the information content of the data.

*4) The 1st panel of*
Figure 6
*is very confusing, given that the “opener” rats would have opened the door at different times – is the claim that each rat’s locomotion decreased immediately after they opened the restrainer, or rather, that it decreased mainly during the last twenty minutes of the session*?

We agree with the reviewers and removed panel 6A.

*5)*
Figure 3
*is confusing. Is the fostered rat that will be tested not depicted in each illustrated litter? What is happening with the first 2 litters that switch from mostly LE to all SD and then back to LE in panel B? Please clarify*.

We apologize for the confusion. We have made several changes to this figure to improve clarity and have also rewritten the figure legend.

[Editors’ note: the author responses to the re-review follow.]

*1) General arousal seems to be a very reasonable alternative to the pro-social interpretation of your results. This explanation needs to be presented as an alternative and discussed*.

We appreciate the reviewers’ efforts to raise multiple alternative explanations, and now include this idea in the Discussion. The reviewers suggest that a non-specific response underlies door-opening behavior. If we understand correctly, the idea is that 1) rats from familiar strains elicit more “general” arousal than do rats from unfamiliar strains; and 2) “generally” aroused rats then move more and thereby open the door accidentally. While it is important to ponder every alternative explanation, our data do not support a general arousal mechanism. Further analysis shows that non-openers tested with unfamiliar strains were as active before door-opening as openers (new Figure 8). These data indicate that general arousal cannot account for differential door-opening.

That being said, we do not doubt that the different conditions elicit different states of arousal for the free rat. Yet, we would not term this “general arousal”. Rather we perceive it as a state that combines arousal with affective valence. This form of specific arousal is elicited by the distress of another rat.

Finally, the general arousal hypothesis would suggest that rats are not acting in a goal-directed manner when they open the restrainer. We have previously demonstrated that door-opening is a goal-directed behavior. For the benefit of reviewers who may not have had a chance to read the previous publication (Ben-Ami Bartal, et al. 2011), we found that rats demonstrate a classic learning curve for door-opening latencies. In fact, door-opening is hard, and rats have to work at it even when the restrainer contains coveted chocolate chips. Rats overcome their initial aversion of the arena center and demonstrate a specific movement pattern that involves circling the restrainer, biting and digging under it, and making multiple attempts to open the restrainer. Rats only cease such restrainer-focused activity after learning to open the restrainer. As the rats learn to open the door, the method for door-opening becomes uniform, and the initial freezing response to sound of the door falling over disappears. Rats did not show these behaviors when the restrainer was empty, contained a toy, or when a non-trapped cagemate was present. As such, general arousal is an unlikely and non-specific explanation for door-opening behavior in this paradigm, much like it would be an uninformative interpretation of a rat locating the underwater platform in the Morris water maze.

*2) The present study doesn’t really contrast an ‘experience’ versus a ‘genetic’ hypothesis*.

We have taken into account the reviewers’ advice, and accordingly de-emphasized the discussion of genetic relatedness in the manuscript. It is true that members of an outbred strain (or stock) are not genetically identical. Yet, it is also true that genetic similarity is higher within strains than between strains. Thus it is safe to say that our findings demonstrate that genetic similarity cannot, on its own, motivate helping behavior, as rats fostered with a different strain do not help members of their own strain.

*What is really interesting is that a rat will treat strangers that look like the rats with whom they were reared as if they were familiars, or least ones that they will preferentially release. The rats single out some cue(s) about their cage mates that they can then apply to strangers and discriminate among them as those that share the cues with cage mates and those that do not. The authors should discuss this and consider what cues might be being used*.

We agree that our finding that rats treat strangers that resemble rats with whom they are familiar, as if they were themselves familiar is really interesting. We have changed our title to better reflect this core finding: “My kind of rat: helping behavior in rats is directed towards groups defined by social experience.”

[Editors’ note: an earlier version of the title, “Pro-social behavior in rats is modulated by social experience”, was chosen for publication.]

As the reviewers state, we find that rats help strangers only when they have social experience with *another* rat of the stranger’s strain. We agree that it would be very interesting to know the identity of the cue or cues upon which rats base a determination of familiarity. Addressing this issue has the potential to illuminate how mammals construct group identity and social attachment, which are considered critical for the development and expression of empathy in both humans and non-human animals. The possible cues that rats could use are now explicitly enumerated in the Discussion. We have no doubt that publication of our fundamental finding will stimulate exactly the type of follow-up studies suggested by the reviewers.

*3) A related concern is that there is no explanation of the observation that the door was opened for cagemates as often as for strangers. Every paper published thus far across species (including mice, rats, monkeys, and humans) shows increased “helping” or “cooperation” or “empathy” for cagemates vs non-cagemates, and there is a lot of theory justifying why this should be the case*.

While there are quite a few studies of the effects of familiarity on pro-social behavior, we do not believe they provide direct evidence regarding rodent helping behavior. Earlier studies on rodent helping did not detail the relationship between the helper and the recipient of help (12; 51). Our previous work only tested cagemates. Thus, to the best of our knowledge there is no published evidence to support the idea that rodents preferentially help individually familiar animals.

With regard to cooperative behavior, [52], 2008) demonstrated cooperative behavior between individually familiar as well as individually unfamiliar animals.

Regarding emotional contagion, there is evidence for affective communication between both individually familiar (35; 31) and individually unfamiliar (10) rodents. Of note, Langford, et al. (2006) studied both familiar and stranger dyads. While they concluded that mice experience emotional contagion for cagemates but not for strangers, the data are open to other interpretations. When mice injected with acetic acid (to induce writhing, a visceral pain behavior) were tested with an uninjected stranger, they showed less pain behavior than did mice tested in isolation. While this result could stem from emotional contagion of the uninjected mouse’s low pain state, the authors interpreted it as stemming from stress analgesia. Further, no comparisons were made between rats tested in the presence of a stranger that was either injected or not injected; comparisons were made only with isolated mice. The error bars in Figure 1 of Langford, et al. (2006) would suggest that a mouse in the presence of a stranger injected with acetic acid shows significantly more writhing behavior than one tested with an un-injected stranger.

With regards to non-rodent species, it should be mentioned that studies in primates and dogs show mixed results regarding a familiarity effect on pro-social behavior, inclusive of evidence for pro-social behavior directed at strangers (e.g., Custance, et al. 2012; Tan, et al. 2013; [59]). Finally, humans routinely express empathy toward strangers such as unknown actors (e.g., [39]) or fictitious characters (e.g., [6] and Batson 2005 for a review of possible underlying mechanisms).

*This suggests that there is something about the paradigm that blocks social cues relevant for individual recognition. If you want to suggest something so out of line with previous results you either need to rule out alternative explanations with more experiments or more honestly acknowledge/discuss why these results differ, especially when the claim is so central to the significance of the paper*.

As detailed above, we do not agree that our results are “so out of line with previous results.” Further, we do not interpret our findings as evidence that there was no individual recognition between the rats studied. First, there was no obstacle to full sensory communication between the free and trapped rat, as the clear restrainer is perforated. Moreover, we have added a new analysis to the manuscript demonstrating differences in how the rats behaved with cagemates and strangers (see below and new Figure 7). We conclude that although rats could distinguish strangers from cagemates, they were motivated to help release both.

Prior to door-opening, rats moved faster when the trapped rat was familiar than when he was a stranger. After door-opening, the velocity was not different. A higher velocity before door-opening is evidence that rats recognize individually familiar vs. individually unfamiliar others. Our interpretation of this result is that familiar and unfamiliar rats elicit either different motivational states or different magnitudes of the same motivational state. Most importantly, despite these differences, the behavioral output of rats tested with familiar or unfamiliar rats from a familiar strain was the same: helping!

In sum, we do not believe that our data stand in direct contrast to consistent published findings that we are aware of. If indeed current theory requires that empathy occurs only between familiar individuals, then perhaps our findings should serve as an impetus to refine theory to better match reality.

*4) The authors dismiss the concerns raised that the releasing may be due to social reward because their previous findings found that releasing can occur in the absence of subsequent contact*.

We apologize if we came off as dismissive of the role of social reward. Of course, social reward is at the root of many social behaviors. If being with other individuals did not elicit positive affect, then it is unlikely that individuals would ever be motivated to interact and form groups. Yet, in our previous study, we found that rats opened the door of a restrainer containing a trapped rat even when they could not play with the released rat. Rats did not open the door of an empty restrainer under these same conditions. These findings led us to conclude that immediate social reward provided by the opportunity to play with a released rat is not required for helping behavior.

*For personally acquainted rats, immediate reward may not be needed because of the past association made between access to an individual and rewarding experiences. Similarly, by showing that unfamiliar rats with some cue identifier marking them as similar to cage mates are preferentially released, it may be simply a two-step association process. Release of a rat with a similar cue may occur because of a past association with social reward. Then, in the present experimental paradigm, the releasing rat may actually obtain the social reward, reinforcing the releasing behavior. That is, in this case the social reward may be important to release a rat that is not familiar, but resembles cage mates. The present findings may be accounted by simple association mechanisms, not requiring higher order processes involving empathy. You should discuss the role of association learning in interpreting your results*.

We largely agree. However, in contrast to the reviewers, we do not view empathy as a higher-order process (often synonymous for some scholars with theory of mind, perspective-taking or understanding of another’s state). Rather, empathy is a basic affective capacity that has deep evolutionary roots in mammalian species associated with affective communication, social attachment, and maternal caregiving (see for instance [19]).

Finally, the principal point of this paper is that motivated helping behavior is different toward rats of familiar vs unfamiliar strains. While we would be remiss if we did not discuss the motivation for helping, we also believe that returning to defend in detail the core conclusion of our previous work would be inappropriate.

*5) “…the most parsimonious explanation is that rats open the restrainer door in order to terminate the trapped rat’s distress”. This sentence should be replaced with something like: “A possible interpretation of the data is that rats open the restrainer door in order to terminate the trapped rat’s distress or in order to terminate their own vicarious arousal.*”

We have revised the sentence to read “Given the greater interest in the trapped rat shown by openers and the lack of aggressive interactions between opener pairs, the authors favor the interpretation that rats open the restrainer door in order to terminate the trapped rat’s distress.”

*6) “The selectivity with which rats engage in helping behavior is further evidence that releasing a trapped rat is an intentional social behavior, congruent with an empathic drive to help some rats, but not others... We conclude that rats form affective bonds through social interactions that motivate empathy and helping. The motivation to help is extended to strangers of familiar strains...” This is an example of how the current version of the paper still concludes that the observed behavior represents rats consciously and intentionally helping others (altruism). As all the reviewers agreed, this conclusion is not warranted*.

This remains our interpretation of the data and we respectfully ask to present our interpretation even if it is at variance with that of “all the reviewers.” We have limited the discussion of possible empathic motivation to the final two paragraphs of the Discussion, and have considerably elaborated on possible alternative explanations so that the thoughtful reader can choose their own interpretation.

It is interesting to note that while other demonstrations of affectively-motivated social behaviors, such as emotional contagion and pair-bonding, are widely accepted in the field, the reviewers find it very difficult to accept the possibility that rodents find the distress of others aversive, and that they experience a positive affect when they have an active role in eliminating the distress of another rat. Yet, basic findings support the conclusion that neural and hormonal mechanisms involved in empathy are evolutionarily conserved across species and that the same affective influences that operate in rodents also drive helping in other mammals, including humans.

*7) Please cite the*
[52]
*paper on how social experience modulates generalized reciprocity in rats, and modify the relevant language in the abstract and discussion to give proper credit to this prior study. The generalized reciprocity mechanism discussed by*
[52]
*may be a useful explanation for the social experience-modulated door-opening behavior described here as well*.

This paper was and is cited in the Introduction. Yet how that study would explain our findings is unclear to us. Rutte and Taborsky found that rats were more likely to give food when they were given food before, a result that the authors interpreted as reciprocated cooperative behavior. In our paradigm, the free rats were never trapped themselves, and there was no indication to them that they would ever require the other rat to release them.

At the risk of perseverating, we want to point out once again that the results of the present paper are novel and exciting because they demonstrate that motivated helping behavior is different toward rats of familiar vs. unfamiliar strains and begins to illuminate the biological basis of group identity. Returning to a full discussion of data published previously digresses from this exciting point.